Denouncing the use of field-specific effect size distributions to inform magnitude

Panzarella Emily emilyp99@my.yorku.ca
Beribisky Nataly
http://orcid.org/0000-0002-9247-497X Cribbie Robert A.
Department of Psychology, York University , Toronto, Ontario , Canada
Gray Andrew
Electronic publication date: 2021 Jun 14
Publication date: 2021
Volume: 9
Electronic Location ID: e11383
Received 2020 Jul 23; Accepted 2021 Apr 9
Copyright: © 2021 Panzarella et al.
Copyright year: 2021
Copyright holder: Panzarella et al.
License: This is an open access article distributed under the terms of the Creative Commons Attribution License, which permits unrestricted use, distribution, reproduction and adaptation in any medium and for any purpose provided that it is properly attributed. For attribution, the original author(s), title, publication source (PeerJ) and either DOI or URL of the article must be cited.
License URL: https://creativecommons.org/licenses/by/4.0/

Keywords: Effect sizes, Distribution of effect sizes, Cohen’s d, Pearson’s r, Effect size magnitude, Practical significance, Statistical significance

Funding: The Social Sciences and Humanities Research Council of Canada 435-2016-1057 This work was supported by The Social Sciences and Humanities Research Council of Canada (435-2016-1057). The funders had no role in study design, data collection and analysis, decision to publish, or preparation of the manuscript.

==============================
An effect size (ES) provides valuable information regarding the magnitude of effects, with the interpretation of magnitude being the most important. Interpreting ES magnitude requires combining information from the numerical ES value and the context of the research. However, many researchers adopt popular benchmarks such as those proposed by Cohen. More recently, researchers have proposed interpreting ES magnitude relative to the distribution of observed ESs in a specific field, creating unique benchmarks for declaring effects small, medium or large. However, there is no valid rationale whatsoever for this approach. This study was carried out in two parts: (1) We identified articles that proposed the use of field-specific ES distributions to interpret magnitude (primary articles); and (2) We identified articles that cited the primary articles and classified them by year and publication type. The first type consisted of methodological papers. The second type included articles that interpreted ES magnitude using the approach proposed in the primary articles. There has been a steady increase in the number of methodological and substantial articles discussing or adopting the approach of interpreting ES magnitude by considering the distribution of observed ES in that field, even though the approach is devoid of a theoretical framework. It is hoped that this research will restrict the practice of interpreting ES magnitude relative to the distribution of ES values in a field and instead encourage researchers to interpret such by considering the specific context of the study.

Introduction

An effect size (ES) is defined as a quantitative representation of an event which assesses a research question, often reported alongside a p-value (Kelley & Preacher, 2011). Whereas p-values only assess whether or not an effect is statistically significant, an ES provides valuable information regarding the strength and direction of relationships among variables. This information is conveyed through an interpretation of an ES value’s magnitude. The recommended approach for interpreting the magnitude of effects is to consider the context of the research; however, the method of choice for interpreting the magnitude of an ES is often published benchmarks, such as those proposed by Cohen (1988). According to Google Scholar (2021), Cohen’s book, which proposes (small, medium, and large) cut-offs for most popular ES measures, has been cited almost 200,000 times!

Recently, researchers have proposed interpreting the magnitude of an ES by considering where it falls relative to the distribution of an observed ES within the specific field in which the work was conducted (e.g., Hemphill, 2003). This approach involves establishing unique benchmark values for interpreting ES magnitude based on the observed ES from sets of studies carried out in a specific field. However, this practice completely ignores the context of the study, which is a critical component when evaluating the meaningfulness of effects. Although some might argue that using the distribution of ES values in a field is taking context into account, we completely disagree with this contention. The context of the study considers factors or circumstances directly relevant to the research question(s), such as prior research, setting, participants, methods, and analyses. In contrast, the benchmark approach based on observed ESs necessarily considers only a general field or subfield.

In this study, we systematically review the popularity of using the observed ES distribution to interpret ES magnitude in psychological research. Further, we discuss the limitations of the approach and provide recommendations for interpreting ES magnitude in a meaningful and appropriate manner.

Reporting effect size

An ES can offer information about the calculated value of effects, the direction of effects, the magnitude of effects, and relevant interpretations of effects (Nakagawa & Cuthill, 2007). Organizations such as The American Psychological Association (APA), Washington, DC, USA, The American Educational Research Association (AERA), Washington, DC, USA, The Society for Industrial and Organizational Psychology (SIOP), Bowling Green, OH, USA, and The National Centre for Education Statistics (NCES), Washington, DC, USA, have released official statements regarding the importance of incorporating ES values into research (Kelley & Preacher, 2011). In turn, almost all journal editors have required that an ES value, and its corresponding confidence interval, be reported alongside, or instead of, p-values and null hypothesis significance test results. A good ES value should be scaled accordingly given the question of interest, be accompanied by a confidence interval, and be asymptotically unbiased, consistent, and efficient (Kelley & Preacher, 2011).

An ES measure can be expressed in either unstandardized or standardized format, with the former representing magnitude in the units of the study variables and the latter representing magnitude in generic units. Examples of unstandardized ES measures includes mean differences and regression coefficients, whereas examples of standardized ES measures include standardized mean differences, correlation coefficients, standardized regression coefficients, and odds/risk ratios (Kelley & Preacher, 2011; Rosenthal, 1994). Standardized mean differences are often categorized as part of the d family of ES measures (e.g., Cohen’s d, Hedges’ g: a bias-corrected version of Cohen’s d, Glass’ Δ: a version of Cohen’s d which uses a single group’s standard deviation), whereas correlation coefficients are often categorized as part of the r family of ES measures (e.g., Pearson’s r, partial/semi-partial r, multiple R, biserial/point-biserial r). Although d was designed as a measure of standardized mean difference and r was designed as a measure of correlation, these measures can be converted between one another in order to facilitate comparisons across settings and designs (e.g., d to r, r to d, odds ratio to d, etc.).

As constructs become more abstract, if they are not measured reliably, standardized ES measures are at risk of increased measurement error (Hedges, 1981). Increased measurement error results in lower standardized ES values (Burchinal, 2008). Although there are adjustments that may be used for reliability that may produce potentially better estimates of ES (Baguley, 2009), there is no substitute for better measurement and ongoing validation of the construct itself (Flake, Pek & Hehman, 2017).

ES values results from the application of a relevant ES measure to given data (Kelley & Preacher, 2011). For example, the correlation between two measures can be expressed with an ES value of r = 0.78. Alternatively, an unstandardized mean difference between two groups on the number of cookies consumed can be expressed with an ES value of, for example, two cookies. However, interpreting an ES value’s magnitude involves combining the numeric value of an ES with the context in which it was computed. Although it is not difficult to report standardized and unstandardized values, the burden is placed on the research to explain their meaning given the context of the study by interpreting ES magnitude (Kelley & Preacher, 2011).

Meaningful contextual interpretations

Interpreting an ES within a research context can be difficult. Asking a statistician whether or not a given ES is important or meaningful, is, in all likelihood, much less useful than posing the same question to a subject matter expert (Anderson, 2019). As Anderson (2019) states, what implications an ES has to a field requires a thorough understanding of the utilized variables and their corresponding measures, as well as the extent to which the finding reflects what is expected by a theory. Other factors that can be valuable in providing meaningful interpretations include how the results situate relative to studies evaluating the same hypothesis(es), the potential impact of the results on individuals, families, policies, and resources (e.g., financial, labour, time) required for implementing recommendations. In other words, subject matter experts are in the best position to comment on extended implications of a finding in real life.

For example, consider a fictional example of interpreting an ES value within context in basic science for research on productivity and focus. Assume a researcher is judging the effect of a novel productivity exercise’s effect on focus when completing a task, and finds that the exercise resulted in an average increase of 15 min of focus per day per employee. Consider the following questions:Is an average increase of 15 min of focus per day after the exercise important?

What are the real-life implications of 15 min of extra focus time?

How long are people focused on average without any intervention?

How much variability is there in the change in focus across employees?

How does the focus increase from this particular exercise compare to other productivity exercises?

Is the focus task used in the exercise relevant to the actual tasks of the employees?

Is the sample used in this study representative of the population of interest?

How does the population sampled from the study translate to other populations?

Is 15 min of extra focus time large enough relative to the cost of introducing the productivity exercise?

These are just a small subset of questions that can be used to address an ES value’s magnitude within the context of the study; further, relevant questions can only be raised and addressed by those with knowledge of the theoretical content and the finding’s real world implications. In other words, these questions cannot readily be answered by a statistician removed from the research project, but instead should be illuminated upon by psychology researchers who specialize in focus and productivity, as well as other stakeholders and experts in education and workplace management.

It is worth noting that in this example, and extending to other areas of the paper, we are interpreting ES values at the population level. However, there could be important and meaningful differences at the individual level that are also worth exploring. For instance, perhaps in this example, an individual who was not able to focus on any task was suddenly productive for 15 min per day (as opposed to zero) following the intervention. This individual ES scenario might be more meaningful than increasing the productivity of a person who was focused for 5 h per day prior to the intervention, but focused for 5 h and 15 min afterwards.

A potential critique regarding in-context interpretations of an ES value may challenge whether a cumulative science can be built at all when this approach is used. Indeed, if interpretations apply to one unique setting only, it is important to speak to how different ES values may be combined and meta-analyzed within a broader setting. We argue that in-context interpretations of an ES value’s magnitude actually facilitates the building of a cumulative science much better than approaches that use benchmark or cutoff values. If researchers define ES magnitude using in-context interpretations, it is much easier for another researcher reviewing the literature to ascertain which studies may be combined and meta-analyzed because of explicitly shared contextual interpretations. Moreover, the resulting estimated effect from a meta-analysis can be interpreted, and therefore expand on in-context interpretations from studies addressing research questions that are identical in their intent and purpose.

In contrast, interpretations of an ES such as “large” and “small” are in danger of masking critical contextual and study-specific information (Pek & Flora, 2018). For example, a researcher combining studies which use the benchmark interpretation of an ES value may believe that they are combining information that is similar because it is “context free.” In actuality, it may be that contextual information could reveal that studies with similar magnitudes differ greatly in terms of their practical significance (highlighting again the limitations of using cut-off values).

Although we believe the contextual approach to be the most useful way to interpret an ES value, this is often not the approach taken by researchers (Manolov et al., 2016). The two approaches we describe below are the traditional cutoff approach and the observed ES distribution-based approach.

Traditional/Popular cutoffs for ESs

Cohen’s d and Pearson’s r are both well-known and popular ES measures among psychological researchers, as they are based on well-known statistics (i.e., standardized mean difference, correlation). Popular ES guidelines for Cohen’s d associate a small effect with an ES value that is between d = 0.2 and d = 0.5, a medium effect with an ES value that is between d = 0.5 and d = 0.8, and a large effect with an ES value that is greater than d = 0.8. (Cohen, 1988). Similarly, popular guidelines for Pearson’s r associate a small effect with an ES value that is between r = 0.1 and r = 0.3, a medium effect with an ES value that is between r = 0.3 and r = 0.5, and a large effect with an ES value that is greater than r = 0.5 (Cohen, 1988). While quantifying the magnitude of an effect is important, ES magnitude is recommended to be interpreted within the context of the specific study (Kelley & Preacher, 2011; Pek & Flora, 2018; Richard, Bond & Stokes-Zoota, 2003). Accordingly, when introducing popular cutoffs in his 1988 book, Cohen states that they are “recommended for use only when no better basis for estimating the index is available” (p. 25).

Interpreting effect size magnitude via field-specific effect size distributions

Recent methodological studies in psychology have expressed that the ES benchmark values for Cohen’s d and Pearson’s r are unrealistic based on the observed distribution of ES values in their specific field (e.g., Brydges, 2019). While some researchers convey that the current threshold values are set too low (Morris & Fritz, 2013; Welsh & Knight, 2014), other researchers claim that such values are set too high, making large ESs too difficult to achieve (Hemphill, 2003; Bosco et al., 2015).

The notion that ES benchmark values are unrealistic relative to observed ES distributions in a field was first explored in a study conducted by Hemphill (2003), who considered the distribution of Pearson’s r across 380 studies that discussed psychological assessment and treatment. Hemphill aimed to “extend Cohen’s benchmarks by deriving empirical guidelines concerning the magnitude of correlation coefficients found among psychological studies” (p. 78). Notably, the distribution of observed ES values explored by Hemphill was not synonymous to a theoretical probability distribution (e.g., t distribution). Instead, Hemphill obtained benchmark values by gathering observed ES values from different studies from psychological assessment and treatment, combining them into a single distribution.

All ES values reported were either presented as correlation coefficients or were converted from Cohen’s d to Pearson’s r to compare observed ES value distributions. Finally, values were divided equally into lower, middle, and upper thirds. Hemphill found that the lower third contained ESs between r = 0 and r = 0.2, the middle third contained ESs between r = 0.2 and r = 0.3, and the upper third contained ESs above r = 0.3. These results led Hemphill to conclude that correlation values less than 0.2 should be considered a small effect, values between 0.2 and 0.3 should be considered a medium effect, and values larger than 0.3 should be considered a large effect, to reflect such distributions.

Furthermore, if we convert these correlation values to a standardized mean difference metric, d values less than 0.4 should be considered a small effect, values between 0.4 and 0.6 should be considered a medium effect, and values larger than 0.6 should be considered a large effect.

Hemphill adopted the observed ES approach as an alternative method for interpreting ES values. For instance, Hemphill noted that his findings “suggest that the value Cohen used to represent a large correlation coefficient occurs somewhat infrequently in many key research studies in psychology and that a lower value might be warranted in some instances (p. 79)”. Accordingly, to interpret ES values using this approach and using Hemphill’s distribution as a reference, a researcher would compare their own observed ES values relative to the benchmarks specified from the 280 psychological assessment and treatment studies in order to obtain a “small”, “medium” or “large” label (to see a demonstration of how this method works with simulated data see https://mfr.ca-1.osf.io/render?url=https://osf.io/34gf9/?direct%26mode=render%26action=download%26mode=render).

This approach is extremely problematic because practical significance often becomes distorted (Pek & Flora, 2018). For example, if the benchmark value for a “large” effect was set to a relatively small value (such as r = 0.3 in the Hemphill study) in a specific field, there is the potential for researchers to claim meaningfulness even in situations where the effect has no practical significance (Pogrow, 2019). Further, working in a context-free setting, Beribisky, Davidson & Cribbie (2019) found that a correlation of r = 0.3 was the smallest association that was deemed meaningful when participants viewed scatterplots of context-free associations. Thus, there is simply no valid theoretical rationale for creating benchmarks of ES values for researchers in psychology (or any other discipline/sub-discipline) by considering the distribution of observed ES values within these disciplines (or sub-disciplines). Above all, encouraging researchers to compare ES values relative to the distribution of ES within a discipline (or sub-discipline) discourages the practice of interpreting ES magnitude within the context of a specific study and promotes reliance on context-free benchmark values to conclude practical significance. Exclusively relying on context-free benchmarks was strongly advised against by Cohen (1988) in his initial writings on the topic, and countless authors since. We expand on this point below by highlighting two specific problems with the field-specific observed ES distribution approach: (1) equating ES magnitude with importance, and (2) publication bias.

Issue 1: equating effect size magnitude with importance

One of the largest theoretical issues that exists when using the distribution of observed ES values in a given field as a guideline for interpreting whether a study produces a small, medium, or large effect is that the context of the study is ignored. This is of particular concern in two paradoxical cases—when a study produces a numerically small ES that is considered large within context, and when a study produces a numerically large ES that is considered small within context.

Consider the following example where a statistically small ES value would be considered significant given the context of the study. In 1988, a correlational analysis was conducted to seek any potential relationships between the intake of aspirin and the reduced risk of death by heart attack (Meyer et al., 2001). Under typical ES guidelines (e.g., Cohen, 1988), a very small effect of r = 0.02 was concluded. However, researchers appropriately interpreted the ES value’s magnitude to be large given the context of the study for various reasons, most importantly that many lives were saved. In this instance, and several others, small magnitude ES values can hold large practical significance (Dunst & Hamby, 2012).

Similarly, consider an example in which a statistically large ES value would be considered insignificant given the context of the study. An experiment conducted in the 1990s analyzed the academic performance of students in high-poverty schools by implementing a reading intervention program called Success For All (SFA). Using popular ES guidelines (e.g., Cohen, 1988), the researchers concluded a very large effect. However, the results were practically insignificant because there were no meaningful improvements on the academic performance of students (Pogrow, 2019). In this case, what turned out to be a poor surrogate outcome could have had negative consequences (e.g., combining this ES with ESs from more legitimate outcomes, potentially inflating the combined effect size). An independent study was conducted a few years later and corroborated that the approach did not produce meaningful improvements in performance; more specifically, the experimental group was performing poorly academically relative to the control group and the program was terminated (Pogrow, 2019).

Thus, the main theoretical issue with the distribution of observed ES values approach is that it distracts researchers from interpreting ES magnitude by considering the specific context of their research study (Valentine & Cooper, 2003; Simpson, 2018). In other words, the approach insinuates that threshold values alone hold all of the answers about a study’s practical significance. Hemphill (2003) encourages the interpretation of ES magnitude within specific fields by considering the distribution of ESs in that field; however, Cohen (1988), and several others (e.g., Kelley & Preacher, 2011; Pek & Flora, 2018), advise us against using proposed benchmarks as a determinant of practical significance alone and strongly encourage the interpretation of ES magnitude by considering the specific context of the study.

Issue 2: publication bias and effect sizes

Publication bias can be defined as an increased likelihood for an effect that is statistically significant, rather than one that is not, to be published. In psychology, statistically non-significant results have sometimes been met with critiques of low power, and therefore increased likelihood of Type II error (Ferguson & Heene, 2012). Due to this inconsistent treatment of research findings, it has been noted that published ES values tend to be larger or inflated compared to their unpublished counterparts (Vevea & Hedges, 1995; Schäfer & Schwarz, 2019; Bushman, Rothstein & Anderson, 2010). As a further complication, the magnitude of publication bias is inconsistent across disciplines and even sub-disciplines of a given field.

This creates a further difficulty for researchers using the distribution of observed ES values approach when interpreting ES magnitude. Specifically, if a researcher is surveying past research on a given topic or sub-discipline, they will be relying upon incomplete literature with potentially over-estimated ES values. Any classification of “small”, “medium”, or “large” will also be inflated, leading to an incorrect interpretation of ES magnitude with the distribution-based approach.

This study contributes to the literature by defining the distribution of observed ES values approach for ES interpretation and explains the problems associated with this approach. This study’s contributions also involve investigating whether this approach is being commonly used.

In the sections above, we highlighted the theoretical issues with the observed distribution of ES values to interpret ES magnitude. In this section of the paper, we seek to quantify the popularity of this approach by highlighting both articles recommending the approach and articles adopting the approach for interpreting ES magnitude. Although the method may be theoretically problematic, if researchers are ignoring it, then there is no issue. However, if this methodology is gaining popularity, then there is a need to highlight the issues with the approach and provide recommendations for interpreting ES magnitude.

Survey methodology

This study is conducted in two parts. First, articles that proposed the use of field-specific ES distributions to inform magnitude were identified and referred to as primary articles. Second, in order to identify the growth of this approach, articles that cited the primary articles were identified and classified by year and publication type. The first publication type consisted of researchers using the primary articles to directly interpret an ES value’s magnitude in their study. The second type consisted of researchers using the concepts from the primary articles to interpret an ES value’s magnitude, but not the specific cut-offs proposed in the study. The last type consisted of methodological articles.

Part I: identifying primary articles

In addition to the Hemphill (2003) study discussed above, we identified four other studies from the field of psychology that proposed ES benchmark value recommendations based on the observed ES values distribution approach, namely Gignac & Szodorai (2016), Rubio-Aparicio et al. (2018), Lovakov & Agadullina (2017), and Brydges (2019) (Table 1). These are particularly influential because they have been published in prospering sub-fields of psychology, including personality and individual differences, clinical psychology, social psychology, and gerontology. Each paper also cites Hemphill’s publication in an attempt to justify their approaches. Further, most have been published recently, highlighting the current popularity of the method.

Table 1 Field-specific effect size (ES) benchmarks proposed by primary articles to interpret ES magnitude.

Field of study	Type of ES	Proposed ES benchmark values	
Small	Medium	Large	
Original benchmark values Cohen (1988)	d	0.2	0.5	0.8	
r	0.1	0.3	0.5	
Psychological assessment and treatment Hemphill (2003)	r	<0.2	0.2	0.3	
Personality and individual differences Gignac & Szodorai (2015)	r	0.15	0.25	0.35	
d	0.1	0.3	0.5	
Clinical psychology Rubio-Aparicio et al. (2018)	d	0.249	0.409	0.695	
Social psychology Lovakov & Agadullina (2017)	r	0.1	0.25	0.4	
Gerontology-psychology of aging Brydges (2019)	r	0.1	0.2	0.3	
Note:

Five primary articles propose field-specific ES benchmarks by considering the distribution of Pearson’s r or Cohen’s d in their field. Values of r or d were divided into lower, middle, and upper thirds to create these benchmarks.

Gignac & Szodorai (2016) considered the distribution of Pearson’s r in personality and individual differences across 87 meta-analyses from six field-specific journals. Similar to Hemphill’s method, all ES values were divided equally into lower, middle, and upper thirds. The researchers found that the lower third contained ESs between r = 0 and r = 0.15, the middle third contained ESs between r = 0.15 and r = 0.25, and the upper third contained ESs between r = 0.25 and r = 0.35. Subsequently, r values less than 0.15 should be considered a small effect, values between 0.15 and 0.25 should be considered a medium effect, and values between 0.25 and 0.35 should be considered a large effect. The authors concluded that differential psychologists will “be arguably better served by applying the correlation guidelines reported in this investigation rather than those reported by Cohen (1988, 1992) or even Hemphill (2003), in order to obtain an accurate sense of the magnitude” (Gignac & Szodorai, 2016).

Rubio-Aparicio et al. (2018) considered the distribution of Cohen’s d in clinical psychology from 54 meta-analyses. The authors recommended benchmark values for researchers in clinical psychology such that between d = 0 and d = 0.249 should be considered a small effect, between d = 0.249 and d = 0.409 should be considered a medium effect, and between d = 0.409 and d = 0.695 should be considered a large effect (Rubio-Aparicio et al., 2018). They also state that “this classification should be used instead of Cohen (1988) proposal, for the interpretation of the standardized mean change values in the clinical psychological context” (Rubio-Aparicio et al., 2018).

Lovakov & Agadullina (2017) gathered 161 meta-analyses among 29 social psychology journals and reported the lower, middle, and upper thirds of Pearson’s r. They recommended ES values between r = 0 and r = 0.1 to be considered a small effect, between r = 0.1 and r = 0.25 to be considered a medium effect, and between r = 0.25 and r = 0.4 to be considered a large effect (Lovakov & Agadullina, 2017), highlighting that Cohen (1988) guidelines were too conservative.

Most recently, Brydges (2019) focused on the distribution of Pearson’s r in gerontology by collecting 88 meta-analyses from 10 field-specific journals. He proposed ES values less than r = 0.1 to be considered a small effect, between r = 0.1 and r = 0.2 to be considered a medium effect, and between r = 0.2 and r = 0.3 to be considered a large effect (Brydges, 2019). Again, Brydges (2019) discusses how Cohen (1988) ES benchmark values were set too high. Similar to the other authors, Brydges (2019) also ignores the fact that Cohen (1988) suggested only using such guidelines when no contextual information is available.

Part II: citations of primary articles

In order to determine if the recommendations of the primary articles found in Part I were being adopted by researchers, we coded articles that cited Hemphill (2003), Gignac & Szodorai (2016), or Rubio-Aparicio et al. (2018) as found on Google Scholar (2020). The research conducted by Lovakov & Agadullina (2017) and Brydges (2019) were excluded from the coding procedure, as they were published too recently to observe any trends within the citations.

First, the team met to derive a coding checklist and discuss the criteria for selecting specific classifications. In order to evaluate the reason behind the citation of one of the three selected primary articles (i.e., to verify that it was to interpret an ES value using field-specific observed ES distributions as opposed to another reason), six coding categories were developed. Publications that used the distribution of observed ES values approach proposed by Hemphill (2003), Gignac & Szodorai (2016), or Rubio-Aparicio et al. (2018) to interpret ES magnitude were coded as Direct Interpretations; papers that used the observed ES value distribution concept from Hemphill (2003), Gignac & Szodorai (2016), or Rubio-Aparicio et al. (2018), but different benchmark values to interpret ES magnitude were coded as Conceptual Interpretations; papers that were of a methodological or theoretical nature were coded as Methodological/Theoretical; and lastly papers that cited a primary article for a reason unrelated to the interpretation of an ES were coded as Irrelevant. Articles that used more than one method for interpreting ES magnitude (e.g., Hemphill’s (2003) approach and Cohen (1988) cutoffs) were coded as Multiple Methods, and articles that could not be accessed were coded as Inaccessible. Articles that used a direct or conceptual interpretation of ES values demonstrate the popularity of applying the field-specific observed ES distributions method to interpret ES values.

Books, dissertations, theses, and non-English articles with no available translations were excluded from the sample.

To establish interrater reliability, the team (the three authors plus five research assistants) used the checklist to code ten articles selected at random that cited any of the three primary articles. The group met to discuss their findings and found a total of 15 discrepancies across 320 items in total (10 articles × 8 coders × 4 items), offering 95.3% agreement among coders. The discrepancies were discussed in order to clarify any issues regarding the coding. Further, a decision regarding any controversial categorizations was made by the authors.

Results

Overall, 426 articles cited Hemphill (2003), 122 articles cited Gignac & Szodorai (2015), and 10 articles cited Rubio-Aparicio et al. (2018), for a total of N = 686 citations. A steady increase in the number of citations to each primary article from peer-reviewed research articles between one year following the publication date of the primary article until 2019 can be observed in Fig. 1. Amongst these peer-reviewed articles, 323 studies (62.5%) were coded as Direct Interpretations, 10 articles (1.9%) were coded as Conceptual Interpretations, 35 articles (6.8%) were coded as Theoretical/Methodological, and 37 articles (7.2%) were coded as Irrelevant. As well, 90 of the above articles (17.4%) were also coded as Multiple Methods, all of which were initially coded as a Direct Interpretation. Finally, 21 articles (4.3%) were coded as Inaccessible (Fig. 2).

Figure 1 The number of citations each primary article has received since their publication date vs. the number of articles published in a journal containing the keyword ‘psychology’.

This histogram represents the number of citations received from the following articles: Hemphill (2003), Gignac & Szodorai (2016), Rubio-Aparicio et al. (2018), Lovakov & Agadullina (2017), and Brydges (2019). The line represents the number of articles that were published in a journal containing the keyword ‘psychology’.

Figure 2 The types of citations received by Hemphill (2003), Gignac & Szodorai (2016), and Rubio-Aparicio et al. (2018).

Articles were coded on the basis of: (1) direct interpretation; papers that used the observed distribution of ES values from a primary article to inform magnitude, (2), conceptual interpretation; papers that used the concept of observed distribution of ES values, but different benchmark values to inform magnitude; (3) methodological/theoretical, (4) irrelevant; papers that cited a primary article for a reason unrelated to the interpretation of an ES value, (5) multiple methods; papers that used more than one method for interpreting ES magnitude, and (6) inaccessible; articles that could not be accessed.

Discussion

Popularizing the use of the distribution of observed ES values within a field weakens the practice of interpreting an ES value’s magnitude within the context of a study by promoting reliance on ES benchmark values, as advised against by Cohen (1988) in his initial proposals. Hemphill (2003), Gignac & Szodorai (2016), Rubio-Aparicio et al. (2018), Lovakov & Agadullina (2017), and Brydges (2019) have supported the distribution of observed ES values approach by proposing field-specific ES benchmarks, which are becoming increasingly influential in psychological assessment and treatment, personality and individual differences, and clinical psychology, subsequently. Our review found that citations to papers proposing field-specific ES benchmarks are growing rapidly within sub-disciplines of psychology as researchers seek suggestions for interpreting the magnitude of ES values. Furthermore, most peer-reviewed articles that cited these primary articles used the field-specific ES benchmarks in the papers to interpret ES magnitude; i.e., they adopted the distribution of observed ES values approach for informing magnitude relative to their own obtained ES (rather than, for example, just mentioning the method). This is extremely problematic, as there is little that can be gained about the meaningfulness of a particular effect, within a particular context, by exploring the distribution of ES values within the narrowly or broadly defined discipline. Moreover, these alternative methods for interpreting ES magnitude discourages the practice of interpreting ES magnitude within the context of the study itself.

To ensure that the observed increase in the number of primary article citations was not a result of an increasing number of journals in psychology overall, we conducted a literature search through Google Scholar (2020) to identify the total number of published articles in journals containing the keyword “psychology” between 2003 and 2019. Interestingly, the total number of articles published in journals containing the keyword “psychology” has remained fairly stable over this time frame (Fig. 1). Thus, the increase in the number of citations received by Hemphill (2003), Gignac & Szodorai (2016), Rubio-Aparicio et al. (2018), Lovakov & Agadullina (2017) and Brydges (2019) do not appear to reflect any overall increase in the number of articles published within the discipline, discounting the reference group problem.

To be clear, we are not arguing that the distribution of ES values in a given field or discipline is of zero value; it is very important to understand the magnitude of ES values in a given discipline and to appreciate how ES magnitude varies within, and across, disciplines. Our point is that researchers should not interpret the magnitude of a particular effect, which occurs within a specific context, by using cut-offs that are derived from an exploration of the distribution of ES values in a broad discipline. For example, when interpreting the magnitude of the relationship between aspirin and heart attacks (Meyer et al., 2001), the distribution of ES values across the discipline at large are trivial compared to the implications of the given study (where the only contextually shared information might be the overarching discipline itself).

One limitation of this study is that we conducted the search in Google Scholar. Although we expect the trend of citations to generalize to other search engines and databases, our findings concern Google Scholar results only.

Conclusions

Using the distribution of observed ES values approach for interpreting ES magnitude distracts researchers from informing magnitude by considering the specific context of their research study. The upwards trend in citations received by the primary articles (e.g., Hemphill, 2003) demonstrates an increase in the attention that this malpractice is earning, and the fact that most citations are for direct interpretations of ES magnitude confirms our hypothesis that researchers are adopting the approach as a crutch for informing magnitude. We hope that this study is valuable in highlighting to researchers both the theoretical issues with the distribution of observed ES values approach, as well as the rising popularity of the strategy; a dangerous combination. Researchers in psychology lack clear guidelines regarding the process of interpreting ES magnitude (e.g., Farmus et al., 2020; Funder & Ozer, 2019), leaving them susceptible to adopting practices that may be of limited value. Psychology is in desperate need of more research regarding best practices for interpreting ES magnitude, but while we wait for these recommendations, journal editors need to encourage researchers to interpret ES magnitude using any contextual information available. Further, it is important that textbook authors, statistics instructors, and the like highlight the importance of not using published benchmark values for interpreting ES magnitude and instead encourage researchers to consider contextual factors. Improving ES magnitude interpretation will not occur overnight, but it is with hope that these recommendations will encourage researchers to consider what factors are valuable when considering the practical significance of their results.

Supplemental Information

Supplemental Information 1 Articles obtained from Google Scholar, who cite the identified primary articles that recommend using field-specific effect size distributions to inform magnitude.

Articles are coded on the basis of: (1) Directly interpretational, use the proposed benchmark values from a primary article. (2) Conceptually interpretational, use the idea but different benchmark values from a primary article. (3) Methodological/theoretical, articles of a methodological or theoretical nature. (4) Multiple methods, use more than one set of benchmark values that include those from a primary article. (5) Irrelevant, cite a primary article for a reason unrelated to the interpretation of effect sizes. (6) Inaccessible, articles that could not be accessed through Google Scholar.

Click here for additional data file.

Supplemental Information 2 Number of Articles in Journals Containing the Keyword “psychology”.

A comparison between the number of published articles that cite the primary authors versus the number of articles published in journals containing the word "psychology".

Click here for additional data file.

Supplemental Information 3 Data simulation of how effect size distributions are used to interpret magnitude.

We open a fake dataset that contains different values of Cohen’s d and Pearson’s r: the standardized mean difference and the correlation coefficient, respectively. This data has characteristics that are similar to Hemphill (2003) but they could just as easily be studies in another sub-discipline or an entire field (e.g., Psychology).

Click here for additional data file.

We would like to thank our research assistants Farwa Abdi, David Civil, David Eldridge, Korina Taguba, and Anh Truong Phuong Nguyen for assisting in the coding process.

Additional Information and Declarations

Competing Interests

Author Contributions

Data Availability

The authors declare that they have no competing interests.

Emily Panzarella conceived and designed the experiments, performed the experiments, analyzed the data, prepared figures and/or tables, authored or reviewed drafts of the paper, and approved the final draft.

Nataly Beribisky conceived and designed the experiments, performed the experiments, authored or reviewed drafts of the paper, and approved the final draft.

Robert A. Cribbie conceived and designed the experiments, performed the experiments, authored or reviewed drafts of the paper, and approved the final draft.

The following information was supplied regarding data availability:

The raw data is available as a Supplemental File.

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
