# Peer review of "Denouncing the use of field-specific effect size distributions to inform magnitude"

_PeerJ, doi:10.7717/peerj.11383_

## Round 0.1 · original submission · Major Revisions

Two reviewers have provided some insightful comments on your manuscript. As you will see, they identify some important limitations of your manuscript as it stands. I think you would be able to address their and my comments, although it may change the focus of the manuscript somewhat. I hope that you will respond to each of their (and my) comments with an explanation of how your manuscript has been changed in response or an explanation of why you have not made any changes.

As Reviewer #1 notes, your manuscript is quite short and there is some disconnect between the theoretical discussion and the survey. I feel that their suggestion of more critically reviewing the primary articles would add value for readers of your manuscript. I agree with your stance that using DES would require justification, but I also agree that you need to establish why DES is flawed, or perhaps more accurately, when it will be flawed. There are two points here that I’ll come back to further below. This reviewer also raises some important questions about when unstandardized ESs, standardised ESs, and DESs might each be useful and this suggests to me the idea of a series of scenarios, guiding the reader through identifying when certain approaches are particularly poorly (or well) suited. If you feel that the answer to the DES question is never, that’s fine, but I think you need to argue this from first principles. As they note, there needs to be some standardisation of the article citations over time (even as a fraction of citations of Cohen). With this in place, perhaps Lovakov & Agadullina (2018) and Brydges (2019) should not be excluded. I don’t think their citations will be interpretable in themselves, but their inclusion can only improve understanding overall trends and you could show the sum of these (again, standardised in some way). Slightly related to this, perhaps kappa could be added to accompany the percentage agreement on Line 268.

Reviewer #2 makes some important connections between your work and the literature and is very helpful in providing references. Of course, you are free to use their suggestions or others of your choosing that provide similar support (or none even). Their points about carefully defining your terms should be useful in guiding the reader through your reasoning. Their point about publication bias seems quite damning for interpreting DESs based on some particular percentile as these biases can vary between fields. I like their suggestion of showing the distribution of ESs, which could of course be bimodal (or even higher multimodal). Finally, the connection they note to MBI resonated for me. While this might not warrant mention in your manuscript, the fear that “important” findings might not be fully appreciated seems to be the common motivator for both MBI and DES.

I’m afraid I share the reviewers' distrust/dislike of pie charts.

One issue that I think needs more attention is the difference between clinical or practical significance at the individual level (where, for example, aspirin has only a small effect on an individual’s risk of a heart attack) and the population level (where health economists may find even small shifts in risk to be important). Related to this, shifts at the population level can be distinguished by where the shift occurs. In clinical settings, we are generally able to interpret unstandardized effect sizes, although it is sometimes important to identify whether one part of the population distribution moves more than another. A reduction of 10mmHg in systolic blood pressure will have more importance if it occurs among those with hypertension than if it occurs among those who are pre-hypertensive, where again it will be more important than if it occurs among those who are normotensive. Alternatively, a reduction in systolic blood pressure of 7% across the board (i.e., greater reductions for those with higher baseline values) would have a greater benefit than then same arithmetic mean effect across the board. Similar points arise for body composition, blood glucose, and other clinical measures. Similarly, modest improvements in mood might be important at the population level despite having little effect on quality of life for individuals in that population, and greater improvements among those with much worse values could be more important than equally-shared improvements.

A second issue that I think is important to acknowledge is the impact of measurement error on standardised effect sizes. The same 10mmHg reduction in systolic blood pressure will have a smaller standardised effect size if the outcome is measured once by a poorly trained nurse immediately when the participant enters the room compared to if the mean of three from an experienced and skilled nurse is used after the participant has been seated for ten minutes. A field with more difficult measures will naturally have lower standardised effect sizes, and this attenuation could be undone if researchers had measures of reliability. As Reviewer #1 notes, what should we do in a field where effect sizes never reach overall “large” magnitudes? It seems odd to deny the possibility of “relatively large” effects in such a field. I would agree with you that using DES creates the risk of “gaming” of ESMs, particularly for research that spans multiple disciplines (not completely unlike journals in multiple categories for JCR, with multiple quartiles and percentiles for which some researchers will be inclined to take the best of the options rather than the category that their article best matches), but using conventional ESMs can also lead to what seem to be incorrect judgments.

I was surprised that the discussion was so brief. What do you consider to be the strengths and limitations of your work? What should, for example, journal editors do in response to your findings? What are the next steps in researching this topic?

Reviewer 1 ·

Basic reporting

no comment

Experimental design

no comment

Validity of the findings

The survey results do not seem to extend or add to the theoretical statements made.

Additional comments

The manuscript titled “Denouncing the use of field-specific effect size distributions to inform magnitude” argues that the interpretation of effect sizes should not use field-specific effect size distributions (DES), but instead provide interpretations within the context of the study. Further, to support the claim that effect size interpretations are increasingly based on field-specific effect size distributions, the authors conducted a survey on how often papers (Hemphill, 2003; Gignac & Szodorai, 2016; Rubio-Aparicio et al., 2017) that promote this perspective are cited and for what purpose.

The paper is written in a clear style, and this perspective is an interesting take toward the interpretation of effects sizes. The paper is relatively short and thus was limited in its theoretical development as well as interpretation of empirical results. Below, I highlight several general points that come across as weaknesses of this current version of the manuscript.

THEORETICAL CONTRIBUTION. The paper makes the strong claim that effect sizes should be interpreted within the context of the study and not according to DES. The primary reason for this position offered is that there is no rationale for the DES approach. Unfortunately, the authors have not shown why there is no rationale for this position when making a declarative statement.

The rationale for collecting effect sizes within a field must have been provided within the key articles of Hemphill (2003), Gignac & Szodorai (2015), and Rubio-Aparicio et al. (2017). It would be helpful to review the motivation of these authors to provide an effect size distribution. Were effect size distributions provided for the interpretation of other effect sizes? Were they provided to obtain a combined meta-effect of the field? Or were these effect sizes collected to provide some basis of power analysis? Effect sizes and their distributions can serve many purposes – meta-analysis, power analysis, and interpretation. It would be helpful to clarify what the original purpose of effect distributions are before narrowing it to the interpretation of effect sizes.

To state that there is no basis to interpret effect sizes using DES seems like general statement that requires substantiation.

The interpretation of effect size magnitudes are straightforward in applied areas of research (e.g., public health, education). Such examples are provided in the arguments against using DES to interpret effect sizes of a study. However, magnitudes in effect sizes in basic research (e.g., persuasion and attitude change, stereotype threat) are more elusive. The difference between applied versus basic research has to do with measurement of the constructs of interest (Blanton & Jaccard, 2006). For instance, the examples about practical significance is straightforward in applied research (e.g., change in blood pressure, number of suicides, number of cigarettes smoked). But, measures of constructs in basic research are not well-established. How “large” is a correlation of r = 0.2 in terms of a mood manipulation on attitude change? Is a d = .5 small or large for the relationship between positive emotions and novel thought-actions? If correlations in a specific area does not ever reach .6 in reality, shouldn’t adjustments be made to the interpretation of the magnitude of the effect size according to what is observed in the field of research? This rationale is consistent with DES and seems to be theoretically based. If DES are not to be used, how does one interpret an effect size with no consensus on what is large vs. what is small? It could be argued here that interpreting effect sizes according to their theoretical range is unhelpful to understand the practical significance of an effect. An empirical example within basic science would be extremely helpful to illustrate the point of how interpretations of effect sizes within the context of the study are to be conducted. There is no consensus on the magnitude of even standardized effect sizes because of the nature of basic research where interest is in the direction of an effect and not on the size of an effect (e.g., see Jones & Tukey, 2000).

If all effect sizes are to be interpreted within the context of the study, then how can a cumulative science be built? Wouldn’t a single effect size be nested within a distribution of relevant effect sizes in a discipline? A deeper theoretical treatment of the interpretation of effect sizes for what purpose (e.g., specific to the study, specific to the field, etc.) can address this concern.
In general, there are many contexts with which effect sizes can be interpreted. More clarity on what is meant by context can help the theoretical development of the ideas presented in the paper.

EMPIRICAL DATA. The survey was conducted to quantify the popularity of the DES approach to interpret effect sizes. How is this purpose related to the theoretical argument that the DES approach should not be used? Stated differently, how does these data add to the argument that the DES approach should not be used?

By focusing on numbers and not frequency, we run into the reference group problem. There are an increasing number of journals, and so the increasing number of citations may reflect the increasing number of publications and not an increase in the DES approach. To show that DES is increasing, it could be helpful to show that the recommended approach to interpret effect sizes is decreasing or remaining constant. If not, then the empirical results provide little evidence to support the argument being made in the paper.

Additionally, articles citing primary articles are categorized into direct interpretations, conceptual interpretations, theoretical/methodological and irrelevant. How do these categories help with the theoretical argument made in the paper?

Minor point:
A bar chart for Figure 2 can allow readers to better make comparisons between the size of the categories.

References
Blanton, H., & Jaccard, J. (2006). Arbitrary metrics in psychology. American Psychologist, 61, 27–41. https://doi.org/10.1037/0003-066X.61.1.27
Jones, L. V., & Tukey, J. W. (2000). A sensible formulation of the significance test. Psychological Methods, 5, 411–414. https://doi.org/10.1037/1082-989X.5.4.411

Reviewer 2 ·

Basic reporting

Pie charts provide a minimal amount of information. It would be better to include the summary information in the text or use an alternative approach to data visualisation.
https://scc.ms.unimelb.edu.au/resources-list/data-visualisation-and-exploration/no_pie-charts

I believe the authors confuse or conflate two distinct issues, the distribution of effect sizes and the categorising or defining of relative effect sizes from available data. I will expound on this issue in the general comments section.

Jacob Cohen’s research is quoted throughout the paper, but none of his work is listed in the references. This seems to be an honest mistake. But, while it is clear one of the references is his power analysis book, it is not clear what the 1992 reference should be. Just in case the authors have missed Cohen’s discussion papers on this topic, I have always found these articles to be very helpful.

https://psycnet.apa.org/record/1995-12080-001
https://psycnet.apa.org/doiLanding?doi=10.1037%2F0033-2909.112.1.155

It is not clear why “Rationale” is a subsection. It consists of four sentences with the first two being transition sentences. It seems the authors would be better served incorporating the rationale within the regular text of the introduction.

Please refrain from defining an abbreviation as a plural, and maybe use an apostrophe s to make it plural, i.e., use “effect sizes (ES)” and “the magnitude of ES’s”. I would also suggest not making any other abbreviations, e.g., use “ES distribution”. I do not feel that ES magnitude has any real meaning since these are usually singular values, i.e., not a vector, and the sign is usually made positive, i.e., direction is often ignored.

Experimental design

No comment

Validity of the findings

No comment

Additional comments

The authors make very strong claims in the title, abstract and text against the use of effect size distributions. My initial impression was they are wholeheartedly wrong which I can demonstrate. However, after a careful read through, I believe their arguments have a lot of merit. Let me explain.

First, there is nothing wrong with effect size distributions. The authors state in the abstract they “are devoid of a theoretical framework”. The distribution of an effect size like, say, Cohen’s d, is related to the t-distribution for normally distributed data. This framework is used to compute p-values and construct confidence intervals, which can be for the mean, mean difference, or standardised mean difference like Cohen’s d. Maximum likelihood estimators, which form the theoretical basis for much of statistical inference, have an approximate normal distribution (by Central Limit Theorem) and can be used to provide similar information.

Hedges did a good job explaining the standardised mean difference distribution, but there are other really good sources out there.

https://doi.org/10.3102%2F10769986006002107

The term “distribution” can mean a probability distribution (as discussed above) but can also apply to a sample of observations. The authors need to clearly delineate these two.

Note that Cohen’s d is a “biased” ES for delta, the true ES. So, the statement on line 92 is not correct unless using Hedge’s bias adjustment for Cohen’s d, or argue for asymptotic unbiasedness.

After a careful reading, I believe the authors are not advising against ES distributions, but instead the somewhat arbitrary use of relative effect sizes. This needs to be made clear in the manuscript and I strongly urge the authors to clearly define these issues and use more standard terminology. Kelley & Preacher (2012), cited by the authors, do a good job discussing the different ways “ES” can be viewed. Others have used this framework to define ES in three ways – (1) ES measure, (2) the observed ES, and (3) relative ES. I believe a fourth use, ES distribution, could also be defined.

https://www.tandfonline.com/doi/abs/10.1080/03610926.2015.1134575

It is the use of the third definition, relative ES, by some authors that the authors are railing against, and I agree with them. Other authors have collected ES’s through systematic reviews which they divide into equal thirds which are labelled as “small”, “medium” and “large” relative ES’s. This approach makes the very strong and unverifiable assumptions that all ES’s are non-zero and they are important in some sense. This is tantamount to moving the goalposts after the kick is made, and the authors are correct in “denouncing” their use.

There will always be a desire to declare an observed ES as large or small. Cohen argued for “operational” ES as medium if “likely to be visible to the naked eye of a careful observer”, small as “not so small to be trivial”, and large as above a medium ES the same distance as small to medium. Without more information about a problem, Cohen’s relative or operational ES’s can be argued for on this basis and has made them very popular over the years.

Cohen was not a big fan of his own approach, but his reasoning has one huge benefit over the “three equal quarters of observed ES” approach. Cohen’s relative ES’s are independent of the observed ES’s. No goalposts are being moved and not all ES’s are important. At a bare minimum, relative ES’s should be chosen to allow for null effects with no constraint of equal numbers of small, medium, and large ES’s. Also, some consideration should be given to the consequences of mislabelling ES (e.g., is labelling a tiny ES generated from an ES distribution with 0 mean as “medium” going to provide misleading information for those in a research area?).

Another issue with creating relative effect sizes from published ES’s is publication bias. We often have a truncated view of observed ES’s since it has been well established that “significant” (i.e., p<0.05) studies are more likely to be published than negative results (i.e., p>0.05).

Can the authors summarise the results visually? I think their points would be clearer if readers could view the distribution of observed ES’s in a histogram, boxplot or other figure.

This paper reminded me of discussions on magnitude-based inference, which has similarly been used to argue that seemingly unimportant results are really “important” based on speculative methods. There have likewise been many critics of this approach.

https://pubmed.ncbi.nlm.nih.gov/25051387/

---

## Round 0.2 · Minor Revisions

One of our hard-working reviewers has made a couple of small suggestions, which I’ll leave you to consider and give you the opportunity to make changes in response to, if you wish. I’ll take this opportunity to also make some small suggestions myself as I anticipate being able to accept your next version of the manuscript (providing no new issues emerge of course) and, in this case, this will facilitate the finalising of the manuscript. My suggestions, like those of the reviewer, should be seen as points to consider, rather than requirements.

Line 66: Perhaps “…regarding the strength AND DIRECTION of relationships…”? (You’re talking about ESs rather than their magnitudes at this point.) This would lead to a small edit of the following sentence as well. This could also lead to changing Line 88 to “…the calculated value, INCLUDING DIRECTION, of effects…”.

Line 71: Could you provide a source for this (with a date as the number will increase over time)?

Line 88: I wonder if this should be “An ES CAN offer information…”.

Line 95: Perhaps “…or instead of, P-VALUES FROM null hypothesis significance tests…” as I think you’re referring more to the p-values rather than just reject/fail to reject.

Line 133: Perhaps the plural “policies” to match the plurality of the other items and recognise that a study might inform multiple policies as well as the collective policy of a government, for example.

Line 177: “identical” might be an overly high bar since I’m not sure of many cases where studies do not vary by time, space, or population in a way that creates (at least subtly) different research questions. Perhaps “…addressing the identical, FOR ALL INTENTS AND PURPOSES, research question…” or similar?

Lines 191–192: Technically, and pedantically, I think, the wording here would either (inclusive between) include 0.5 twice or (exclusive between) not include 0.8 at all. The same seems to apply to Lines 193–194 for 0.3 and 0.5 respectively. Note that you avoid this issue on Lines 220–221 with slightly different wording (presumably in response to the lack of a lower limit for the lowest label). See also Lines 333–336, 343–345, 351–352, and 356–358.

Line 242: Should this be “(or any other discipline/SUB-DISCIPLINE)” given the introduction of sub-disciplines on the following line?

Line 267: I wonder if it’s worth highlighting to the reader that the issue here is one of validity and that apparently the wrong thing was measured in this study, e.g. “…given the context of the study, IN THIS CASE AS THE LARGE EFFECT SIZE WASN’T FOR A VALID MEASURE OF THE INTERVENTION’s SUCCESS…” You could also say “was for a surrogate outcome” or “was for an intermediate outcome” instead. At least part of your point here, I think, is that this large surrogate effect size could be standardised and then unthinkingly combined with a very small standardised effect size for actual academic achievement, leading to a pooled “medium” effect size. Alternatively, you could say on Line 275 something like: “In this case, what turned out to be a poor surrogate outcome could have been combined with valid outcomes, where the distinction could be obscured by using standardised ES.”

Line 289: I don’t think you need the “the” before “likelihood” here.

Line 347: The original article seems to say: “This classification should be used instead of Cohen’s (1988) proposal, for the interpretation of the standardized mean change values in the clinical psychological context.”, i.e. without the pluralisation of “change” in your version. Note that Rubio-Aparicio, et al. was published in its final form in 2018, not 2017 (Line 553).

Line 363: This heading seems to be the only one under Line 362 and so seems unnecessary.

Line 366: Could you add the date the Google Scholar searches were run here?

Line 379: The font seems to change here (“lastly”).

Lines 395–396: Sorry if I’m misunderstanding this but 426+122+10 would be 558, not 686, so I’m assuming that there are (a fair number of) secondary articles citing multiple primary articles. Perhaps it would be useful for readers to add this information on Line 396, e.g. “for a total of N = 686 citations in ?517? articles.” [based on 323/0.625 from Line 399]? However, if the denominator was 517, that would make the 21/517 on Line 403 for me 4.1% rather than the stated 4.3%, so perhaps I’m wrong here?

Line 432: I’d be slightly more cautious here and say: “…do not APPEAR TO reflect any…”

Line 444: Perhaps “…other search engines AND DATABASES,…” to cover Scopus, WoS, etc.

Line 502: There seems to be a spurious indentation here.

Line 507: This is different to your usual DOI presentation. https://doi.org/10.31234/osf.io/nvczj works but https://psyarxiv.com/n5rc7/ seems to have been removed?

Line 516: Another non-standard URL for the DOI. See also Lines 518–519, 536, and 577. A few articles don’t have DOIs listed but these do exist, e.g. http://doi.org/10.1037/a0022658 for Kelley and Preacher (Lines 525–526).

Lines 520–522: The font seems to change here.

Line 528: Note their online publication last week: https://doi.org/10.1002/ejsp.2752.

Figure 1: There is presumably a separate scale (a second y-axis) for the articles with the keyword. Could this be added to the right hand side of the figure? “The passing line” is unusual wording for me, perhaps just “The line”?

Reviewer 2 ·

Basic reporting

No comment

Experimental design

No comment

Validity of the findings

No comment

Additional comments

Thanks for thoroughly addressing my earlier concerns.

I would still like a visual display of the data but do not want to hold up publication. The authors are correct that I would like to see the raw effect sizes which are then put into thirds or quarters (as per Hemphill and others). It is a shame Hemphill did not supply this data and this is no fault of the current authors. But I do believe this would make it much easier for the authors to explain how effect size distributions work and are used to interpret magnitude. Perhaps the authors could simulate data for demonstrative purposes?

---

## Round 0.3 · accepted · Accept

Thank you for your responses and revisions. I am delighted to accept your manuscript, which I am sure will generate some robust discussion and, I am optimistic, will ultimately lead to better practices around interpreting effect sizes. As a biostatistician who is both amused and bemused when asked to interpret effect sizes, I look forward to being able to point my collaborators to your work.

My only comment is about Figure 1. I had asked “Figure 1: There is presumably a separate scale (a second y-axis) for the articles with the keyword. Could this be added to the right hand side of the figure?” I appreciate that one of the purposes of the figure is to show that the number of “psychology” articles is stationary, but as a reader, I think I would be puzzled as to what the line represents in a numerical sense. If this line is equivalent to, e.g. 1 unit on the y-axis (normally citations) = XXXX such articles published, this value (XXXX) could be easily added to the legend as an alternative to the addition of a second y-axis. My apologies if I’m overlooking this information somehow.